# Molecular basis of enzymatic nitrogen-nitrogen formation by a family of zinc-binding cupin enzymes

Guiyun Zhao[1,2,5], Wei Peng[3,5], Kaihui Song[2], Jingkun Shi[2], Xingyu Lu[4], Binju Wang [3✉] & Yi-Ling Du [1,2✉]

Molecules with a nitrogen-nitrogen (N-N) bond in their structures exhibit various biological activities and other unique properties. A few microbial proteins are recently emerging as dedicated N-N bond forming enzymes in natural product biosynthesis. However, the details of these biochemical processes remain largely unknown. Here, through in vitro biochemical characterization and computational studies, we report the molecular basis of hydrazine bond formation by a family of di-domain enzymes. These enzymes are widespread in bacteria and sometimes naturally exist as two standalone enzymes. We reveal that the methionyl-tRNA synthase-like domain/protein catalyzes ATP-dependent condensation of two amino acids substrates to form a highly unstable ester intermediate, which is subsequently captured by the zinc-binding cupin domain/protein and undergoes redox-neutral intramolecular rearrangement to give the N-N bond containing product. These results provide important mechanistic insights into enzymatic N-N bond formation and should facilitate future development of novel N-N forming biocatalyst.

[1] State Key Laboratory for Diagnosis and Treatment of Infectious Diseases, The First Affiliated Hospital, School of Medicine, Zhejiang University, 310003 Hangzhou, China. [2] Institute of Pharmaceutical Biotechnology, School of Medicine, Zhejiang University, 310058 Hangzhou, China. [3] State Key Laboratory of Physical Chemistry of Solid Surfaces and Fujian Provincial Key Laboratory of Theoretical and Computational Chemistry, College of Chemistry and Chemical Engineering, Xiamen University, 361005 Xiamen, China. [4] Key Laboratory of Precise Synthesis of Functional Molecules of Zhejiang Province, School of Science, Instrumentation and Service Center for Molecular Sciences, Westlake University, 310024 Hangzhou, China. [5]These authors contributed equally: Guiyun Zhao, Wei Peng. ✉email: wangbinju2018@xmu.edu.cn; yldu@zju.edu.cn

Molecules containing nitrogen-nitrogen (N-N) linkages display diversified structures and biological activities and make up a large proportion of clinical drugs[1]. Although hundreds of these molecules have also been isolated from nature (Fig. 1a), the biosynthetic enzymes that are responsible for constructing those unusual N-N bonds, including hydrazine, diazo, and *N*-nitroso moieties, are only starting to be uncovered very recently[2–14]. Among them, the piperazate synthase KtzT is a heme-dependent enzyme that catalyzes N-N cyclization of L-$N^5$-OH-ornithine to give the hydrazine-bearing piperazate, which is a nonproteinogenic amino acid building block for many non-ribosomal peptides[2]. In the biosynthetic route to the cancer chemotherapeutic streptozotocin, an iron-binding metalloenzyme SznF appears to mediate an oxidative rearrangement of *N*-methyl-L-arginine to give an *N*-nitrosourea product[3,7]. Besides, the ATP-dependent ligase CreM and the transmembrane protein AzpL, are found to be involved in the formation of diazo bonds in the biosynthesis of cremeomycin and alazopeptin, respectively[6,12]. Despite that several enzymes from distinct protein families have been related to N-N bonds formation, the catalytic details of these unusual biochemical transformations are largely unknown in most cases.

It has been noticed that a collection of three genes, encoding a lysine/ornithine *N*-hydroxylase, a cupin protein, and a methionyl-tRNA synthase (MetRS) homolog are widely distributed in bacterial species[15]. In some cases, the cupin and MetRS-like proteins are fused into a single di-domain protein. However, it was until recently, the functions of these genes were linked to the biosynthesis of N-N bond containing microbial specialized metabolites through in vivo studies[5,8,16–18]. For instance, the di-domain proteins, consisting of an N-terminal cupin domain and a C-terminal MetRS-like domain, have been identified in the

biosynthetic gene clusters (BGCs) of molecules including s56-p1[5], triacsin A[16], and pyrazomycin (also known as pyrazofurin) (Fig. 1b)[8,17,18]. Preliminary studies based on in vivo biotransformation experiments have shown that Spb40 from the s56-p1 BGC catalyzes hydrazine bond formation between L-$N^6$-OH-lysine and L-glycine[5], whereas PyrN from the pyrazomycin pathway displays alternative substrate specificity, linking L-$N^6$-OH-lysine to L-glutamate to form the product 1[8]. However, due to the lack of in vitro biochemical data, how this family of hydrazine synthases mediates N-N bond formation remains elusive.

In this study, through in vitro biochemical assays, catalytic intermediate characterization, enzyme mutagenesis studies, and computational simulations, we reveal the detailed reaction route and the catalytic mechanism of this unusual N-N bond formation process, which involves a family of zinc-binding cupin proteins/domains. Our results provide important mechanistic insights into the biosynthetic strategies for N-N bond construction in nature.

## Results

**In vitro reconstitution of PyrN-catalyzed N-N bond formation.** To reveal the details of PyrN-mediated N-N bond formation, we set out to reconstitute the PyrN-catalyzed reaction in vitro. We first prepared the N-terminal His-tagged PyrN protein from the *E. coli* heterologous expression system (Supplementary Fig. 1a). Considering that PyrN is predicted to contain a C-terminal MetRS-like domain that shares sequence homology to aminoacyl-tRNA synthetases (AARSs) (Fig. 2a), we incubated the isolated PyrN protein with chemically synthesized L-$N^6$-OH-Lys (2) and L-Glu, in the presence of ATP, Mg$^{2+}$, and glutamyl-tRNA, the latter of which was provided in S30 premix extract (Promega).

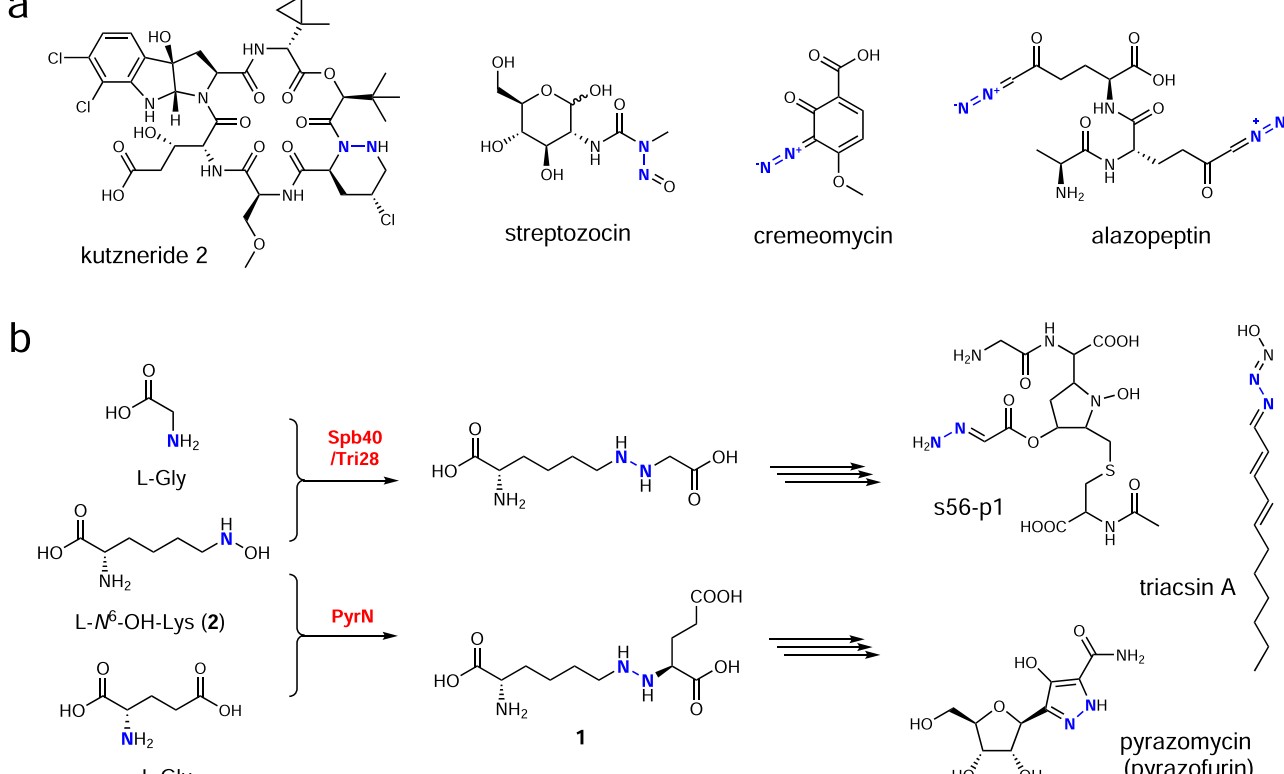

**Fig. 1 Selected natural products containing a nitrogen-nitrogen bond and enzymatic N-N bond formation by a family of di-domain proteins. a** Selected N-N bond containing microbial natural products. **b** Reaction catalyzed by the di-domain enzyme Spb40/Tri28 or PyrN in the biosynthetic pathway of s56-p1, triacsin A, and pyrazomycin, respectively.

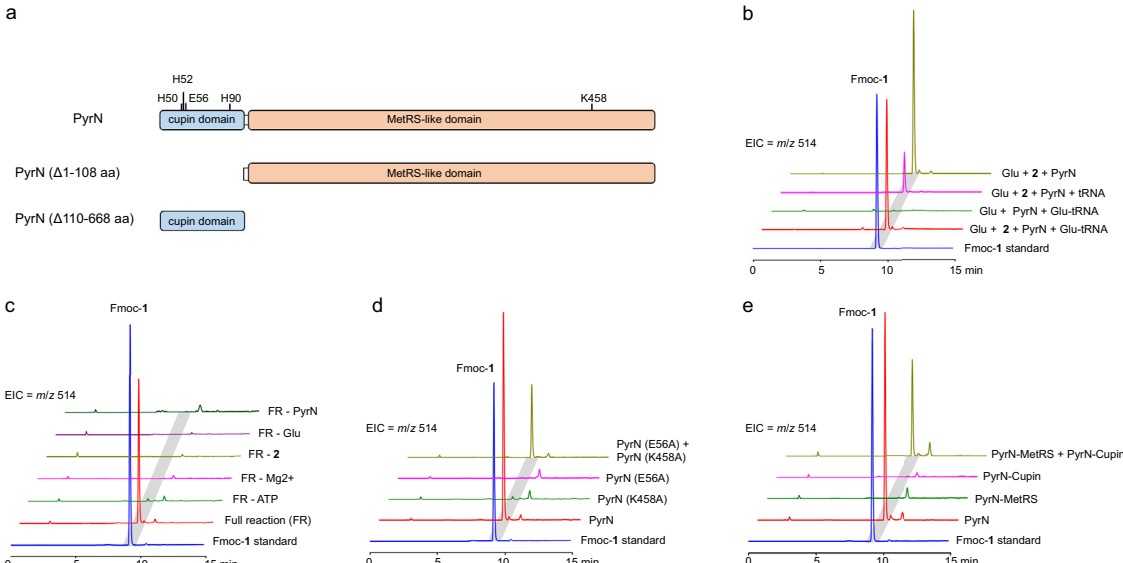

**Fig. 2 In vitro biochemical characterization of PyrN-catalyzed N-N formation. a** Domain organization of the di-domain enzyme PyrN. Putative key residues involved in metal ion and ATP binding were also shown, together with the two truncated PyrN constructs used in the following assays. **b** PyrN-catalyzed tRNA-independent formation of **1**. Extracted ion chromatograms (EIC = $m/z$ 514, $[M + H]^+$ ion for Fmoc-**1**) from the LC-MS analysis of PyrN reaction mixtures were displayed. The different combinations of components contained in the reaction mixtures were indicated next to the corresponding LC-MS traces. **c** In vitro biochemical assays reveal the essential components to the production of **1**. Note: the full reaction (FR) mixtures contain PyrN, ATP, L-Glu, L-$N^6$-OH-Lys (**2**), and MgCl₂ in 40 mM Tris-HCl buffer (pH 8.3). **d** In vitro biochemical assays of PyrN variants. **e** In vitro biochemical assays of truncated PyrN constructs that contain only the MetRS-like domain or the cupin domain, as shown in (**a**).

Moreover, total tRNA mixtures from *E. coli* (Roche) was also used in place of glutamyl-tRNA in a separated assay. LC-MS analysis of the above reaction mixtures after Fmoc chloride (Fmoc-Cl) derivatization, revealed that they both generated a new product with the same retention time and mass signal ($m/z$ 514, $[M + H]^+$ ion) to that of Fmoc-**1** (Fig. 2b), the latter of which was isolated from the culture supernatant of an engineered *E. coli* strain expressing both the lysine-$N^6$-hydroxylase gene *nbtG* and *pyrN*[8], and structurally characterized based on extensive NMR spectroscopic analysis (Supplementary Fig. 2). The N-N linkage in Fmoc-**1** was confirmed by using $^{15}N$ NMR and $^1H$-$^{15}N$ heteronuclear multiple bond correlation (HMBC) NMR analysis, on the Fmoc-**1** sample prepared with L-ε-$^{15}N$-lysine and L-$^{15}N$-glutamate as feeding precursors (Supplementary Fig. 2g, h). Together, these results demonstrated that **1** was successfully produced in the above in vitro reaction mixtures.

Next, we test the tRNA-dependence of PyrN-catalyzed N-N bond formation. We removed glutamyl-tRNA/total tRNA mixtures from the above in vitro reaction mixtures or added RNase in the assay. Interestingly, **1** was still produced in all these cases, suggesting that the PyrN-mediated reaction is tRNA-independent (Fig. 2b and Supplementary Fig. 3a). To further interrogate the necessity of other components in the above in vitro system, they were individually eliminated from the reaction mixtures, and followed by LC-MS analysis. The results demonstrated that L-Glu, **2**, PyrN, ATP, Mg²⁺ are essential to the production of **1** (Fig. 2c).

**The essential roles of both MetRS-like domain and cupin domain in PyrN-catalyzed reaction.** The di-domain organization of PyrN and the previous in vivo studies on Spb40/PyrN, have suggested that the formation of N-N linkage is likely a multi-step process[5,8]. Having established the in vitro reaction condition for the full-length PyrN, we next explore the catalytic role of each domain in vitro. The ATP-dependent nature of PyrN-catalyzed reaction is consistent with the previous in vivo results, which

shows that a point mutation at Lys458 in the putative ATP-binding motif of MetRS-like domain abolished the activity of PyrN[8]. In line with this result, PyrN (K458A) also showed negligible activity in vitro (Fig. 2d). We next interrogate the role of the N-terminal cupin domain (Fig. 2a). The cupin family proteins are known for their ability to bind metal ions for mediating a variety of biochemical transformations[19]. Here, we mutated the four conserved putative metal-chelating residues (His50, His52, Glu56, and His90) in the cupin domain (Fig. 2a), and assayed the activities of the corresponding variants in vitro (Supplementary Fig. 3b). We found that the E56A variant completely abolished the production of **1**, whereas the H50A and H90A variants still retain significant activities (~82 and ~27% relative to the wild-type enzyme, respectively) (Supplementary Fig. 3b and 3c). In addition, the activities of the H52A and the H50A/H90A double variants were severely affected (<3%). Taken together, these results demonstrated that both the MetRS-like domain and the cupin domain are indispensable for PyrN-catalyzed reaction.

Next, we determine whether the two domains of PyrN act independently. We observed that, although both the E56A and K458A variants are inactive on their own, the production of **1** can be restored in a reaction mixture with both variants presented (Fig. 2d). Furthermore, **1** can also be generated upon incubation of the truncated PyrN protein (Δ1–108 aa) that only carries the MetRS-like domain, with the one (Δ110–668 aa) that only contains the cupin domain (Fig. 2a, e). Together, the above results established the essential roles of both MetRS-like domain and cupin domain in PyrN-mediated transformations to afford **1**.

**The catalytic role of MetRS-like domain in PyrN-catalyzed reaction.** We next set out to identify potential reaction intermediate(s) in PyrN-catalyzed N-N bond formation. Based on the sequence homology of PyrN MetRS-like domain to aminoacyl-tRNA synthetases (AARSs), as well as the ATP-dependence of PyrN-catalyzed reaction, we speculated that the MetRS-like domain could catalyze the initial ligation of L-Glu and ATP to

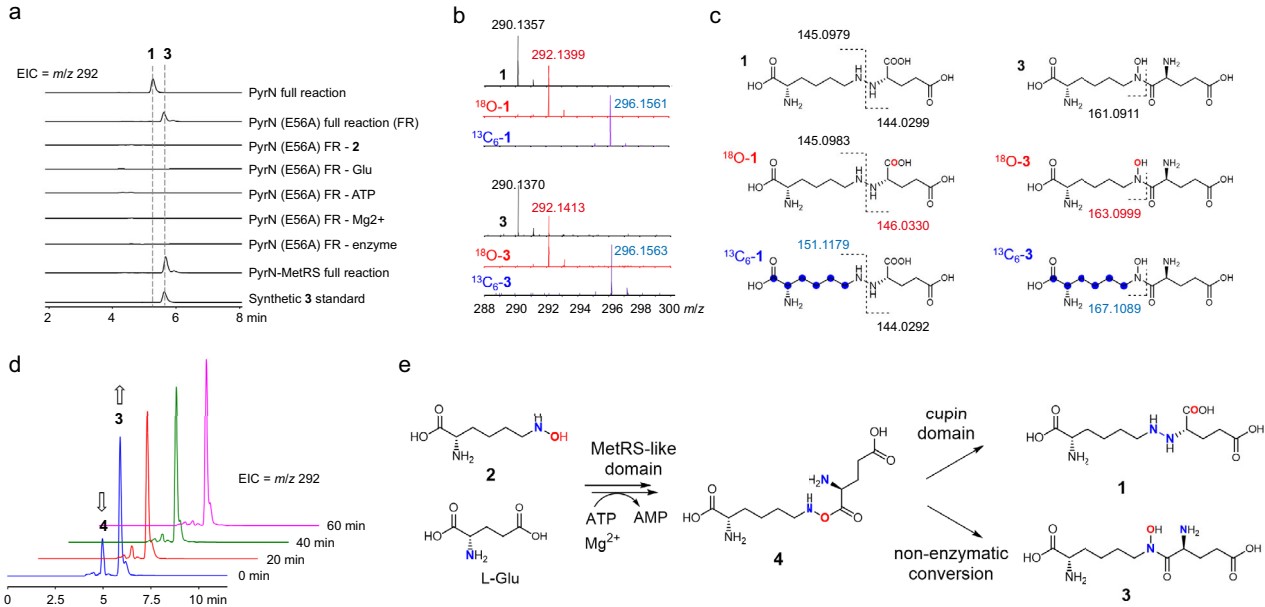

**Fig. 3 Characterization of the reaction intermediate and shunt product from the PyrN-catalyzed reaction. a** LC-MS analysis of the in vitro reaction mixtures of the PyrN (E56A) variant and the truncated MetRS-like domain of PyrN, PyrN MetRS. **b**, **c** LC-HR-MS/MS analysis of **1** and **3** in the reaction mixtures containing isotope-labeled substrates. Selected diagnostic fragment signals are displayed in (**c**). **d** Detection of an unstable reaction intermediate **4** by LC-MS analysis. **e** Proposed reactions routes of PyrN-catalyzed N-N bond formation based on this study.

afford glutamyl-AMP, which is in analogy to the first half reaction of AARSs[20]. The subsequent combination of glutamyl-AMP with **2** could either be mediated by both the MetRS-like and cupin domains, or the cupin alone. To unveil the reaction details, we first focus on the PyrN (E56A) variant, which contains a functional MetRS-like domain, and has a higher protein expression level compared to that of the truncated MetRS-like domain alone (Supplementary Fig. 1a). Incubation of the PyrN (E56A) variant enzyme with L-Glu, **2**, ATP, and $Mg^{2+}$ afford a product (**3**) that displays the same mass signal ($m/z$ 292) but different retention time to that of **1** (Fig. 3a). Moreover, the production of **3** is also strictly dependent on all the components that are essential for the wild-type PyrN reaction (Fig. 3a), and both these reactions generate AMP (Supplementary Fig. 4). We further found that **3** can also be produced when we substituted the truncated MetRS-like domain protein PyrN (Δ1–108 aa) for the PyrN (E56A) variant protein in the above assay (Fig. 3a). Together, these results suggested that **3** might be the reaction product from the PyrN MetRS-like domain.

To characterize compound **3**, we first prepared [18]O-labeled or [13]$C_6$-labeled substrate L-$N^6$-[18]OH-Lys ([18]O-**2**) and L-$N^6$-OH-[13]$C_6$-Lys ([13]$C_6$-**2**), and included them in the in vitro assays of both wild-type PyrN and its E56A variant. Subsequent LC-HR (high resolution)-MS/MS analysis of the above reaction mixtures demonstrated that, although **3** and **1** share the same molecular weight, their fragment patterns are distinct from each other (Fig. 3b, c and Supplementary Fig. 5). More specifically, the [18]O atom is still attaching to the N6 atom of the lysine structural subunit in **3**, whereas the [18]O atom in **1** has been cleaved from the N6 atom and rearranged into one of the carboxyl oxygens in the glutamate subunit (Fig. 3c). This fragment pattern of **3** is supportive of an amide molecule, $N$-glutamyl-$N^6$-hydroxyl-lysine. To further ascertain the structure of **3**, we chemically synthesized the authentic compound of $N$-glutamyl-$N^6$-hydroxyl-lysine, which displays identical retention time and fragment pattern to that of **3** in LC-MS/MS analysis (Fig. 3a and Supplementary Fig. 6). Together, the above results unambiguously established the structure of **3** as $N$-glutamyl-$N^6$-hydroxyl-lysine.

We next test whether **3** is an on-pathway reaction intermediate. We incubated **3** with PyrN, however, **1** was not detected in the reaction mixture upon LC-MS analysis (Supplementary Fig. 7), indicating that **3** is likely an off-pathway shunt product that might derive from an unstable reaction intermediate. To get access to the direct enzymatic product from the MetRS-like domain, we quickly quench the E56A reaction shortly after the reaction is initiated, through the removal of proteins by ultracentrifugation, and immediately analyze the filtrate by LC-MS. We found that, besides **3**, another small peak (**4**) with the same mass signal ($m/z$ 292) could be detected (Fig. 3d). Time course study revealed that this unstable compound rapidly converts to **3**, with a half-life estimated to be less than 10 min.

It has been known that $O$-acyl hydroxylamine rearranges to $N$-acyl hydroxylamine[19,21]. For instance, in the biosynthetic pathway of antibiotic valanimycin, an unstable ester intermediate with a similar N-O bond nonenzymatically converts to the corresponding amide shunt product[21]. Accordingly, our results strongly suggested that the genuine product from the MetRS-like domain is the unstable ester $O$-glutamyl-$N^6$-hydroxyl-lysine, which rapidly arranges to **3** in the absence of a functional cupin domain (Fig. 3e). This proposal was also supported by the above [18]O-labeling pattern in product **1**, which indicated the presence of such an ester intermediate in the PyrN-catalyzed N-N bond formation (Fig. 3c). A similar putative ester intermediate was also proposed in the Spb40-mediated reaction[5]. We further performed LC-HR-MS/MS analysis on **4**, which was generated in situ by the E56A-catalyzed reaction (Supplementary Fig. 8). The results again supported the structural assignment of **4** as $O$-glutamyl-$N^6$-hydroxyl-lysine. Altogether, our results demonstrated that the PyrN MetRS-like domain catalyzes ATP-dependent condensation of L-Glu and **2** to give **4**, which might subsequently undergo cupin-mediated intramolecular arrangement to afford the N-N-containing product **1**.

**The catalytic role of cupin domain in PyrN-catalyzed reaction.** We next investigate cupin-catalyzed N-N bond formation. Our previous study has suggested that the cupin domain of PyrN

binds zinc ion[8]. Here, we use ICP-MS (inductively coupled plasma-mass spectrometry) to further analyze the artificially truncated cupin domain, PyrN (Δ110–668 aa), revealing ~0.5 equivalent of bound zinc. We then test the metal-dependence of PyrN-catalyzed N-N bond formation by the inclusion of metal-chelating agent 1,10-phenanthroline (OP) or ethylenediaminetetraacetic acid (EDTA) into the in vitro assays. We found that OP could efficiently inhibit the production of 1 in a PyrN-catalyzed reaction (Supplementary Fig. 9a). More specifically, OP only blocks the N-N bond-forming step mediated by cupin domain, as we could still detect the production of 3 (Supplementary Fig. 9b). On the other hand, although EDTA appeared to have no effect on the wild-type PyrN (Supplementary Fig. 9a), it completely inhibits the N-N forming activity of the H90A and H50A variants (Supplementary Fig. 9c), indicating that the metal ion is relatively tightly bound by PyrN, but could be stripped out by EDTA after one of metal-chelating His residues was replaced. Taken together, the above results established the metal-dependent nature of cupin-catalyzed N-N bond formation.

**Identification of a zinc-binding N-N forming cupin enzyme by genome mining.** Our attempts to crystalize either the full-length PyrN or its artificially truncated versions have been fruitless so far, we thus turn to search for naturally occurring standalone cupin proteins that might have the same catalytic activity as that of the PyrN cupin domain. We use a genome mining approach to specifically target putative BGCs that also carry homologs of the $N^6$-lysine hydroxylase gene *pyrM* and the putative saccharopine dehydrogenase gene *pyrL*, the latter of which was suggested to convert 1 to 2-hydrazinoglutaric acid in the biosynthetic pathway of pyrazomycin[8,17,18]. Based on this mining strategy, a putative BGC from strain *Rhodococcus jostii* RHA1 was identified and drew our particular attention (Supplementary Fig. 10a). The standalone cupin protein (NCBI accession number: WP_007299751, renamed here as RHS1) from this BGC has been previously crystallized and shown to be a zinc-binding protein (PDB accession number: 5UQP) (Fig. 4a and Supplementary Fig. 11). However, neither the metabolic product from this putative BGC nor the function of the cupin protein RHS1 were known. We generated an engineered *Rhodococcus jostii* strain that overexpressed its own $N^6$-lysine hydroxylase, the standalone MetRS-like protein, and RHS1. LC-MS analysis of the culture supernatant of this strain revealed the production of 1, supporting that the standalone MetRS-like protein also utilizes 2 and L-Glu as substrates (Supplementary Fig. 10b). To further confirm whether RHS1 is functionally equivalent to the PyrN cupin domain in vitro, we prepared His6-tagged recombinant RHS1 (Supplementary Fig. 1b) and included it in the in vitro assays with the PyrN MetRS-like domain protein or the PyrN (E56A) variant, both of which could generate the unstable product 4 in situ for the subsequent cupin-mediated reaction. LC-MS analysis of the

above reaction mixtures revealed the production of 1, demonstrating that RHS1 has the same N-N bond-forming activity to that of the PyrN cupin domain (Fig. 4b). Moreover, we found that 1,10-phenanthroline (OP) could similarly inhibit the activity of RHS1 in the above reaction, supporting the zinc-dependence of this conversion (Supplementary Fig. 12).

Analysis of the RHS1 crystal structure revealed that the zinc ion is coordinated by Asp63, His65, Glu69, and His103 (Fig. 4a). We individually replaced these residues with alanine and assayed the resulting variants. LC-MS analysis revealed that all these variants failed to convert 4 to 1 (Fig. 4c). We then determined the zinc contents of these variants by using ICP-MS, which showed that both the H65A and H103A variants lost their zinc-binding abilities, whereas the D63A and E69A variants still bind one equivalent of zinc ion (Supplementary Fig. 13a). We further found that all these four RHS1 variants failed to restore their activity even in the presence of exogenous zinc ion (100 μM) in the reaction buffer (Supplementary Fig. 13b). Besides the above four zinc-binding residues, we also mutate residues surrounding the active site cavity. These includes Arg26, Ser28, Leu32, Ile52, Thr60, Val67, Trp71, and Trp119 (Fig. 4a). We observed that, except the three variant genes (I52A, T60A, and W71A) that were expressed in the inclusion body, all the other soluble enzymes could still produce significant amounts of 1 in in vitro assays, indicating that they are not critical for catalysis (Supplementary Fig. 13c).

**Computational studies of RHS1-catalyzed N-N bond formation.** The high instability of substrate 4 makes it impossible to obtain an enzyme-substrate co-crystal structure, we thus turn to computational approaches to gain mechanistic insight into the novel cupin-catalyzed N-N bond formation. On basis of the RHS1 crystal structure, 4 was docked into the active site (Supplementary Fig. 14). Two representative binding conformations (conformation-1 and conformation-2) were selected for further mechanistic study (Supplementary Fig. 14a, b). Molecular dynamics (MD) simulations showed that both conformations are stable (Supplementary Fig. 14c, d). For conformation-1, our QM calculations show that the first N-O cleavage step is endothermic by 41.4 kcal/mol (RC′ → IC1′ in Supplementary Fig. 15), indicating that conformation-1 is highly unfavorable for N-O cleavage reaction and thus can be reasonably ruled out. We then focus on conformation-2, in which the substrate O5 and N1 atoms are coordinated to Zn (Supplementary Fig. 14b). The QM/MM calculated energy profile for the RHS1-catalyzed conversion of 4 to 1, from the MD equilibrated representative structure of conformation-2, is shown in Fig. 5a. In the initial reactant complex (RC), the imino group (N1-H) of 4 forms a strong H-bond with the carboxyl O2 of Glu69, while the NH3 group of 4 is H-bonded to the carboxyl O4 of Asp63 (Fig. 5b and Supplementary Fig. 16a).

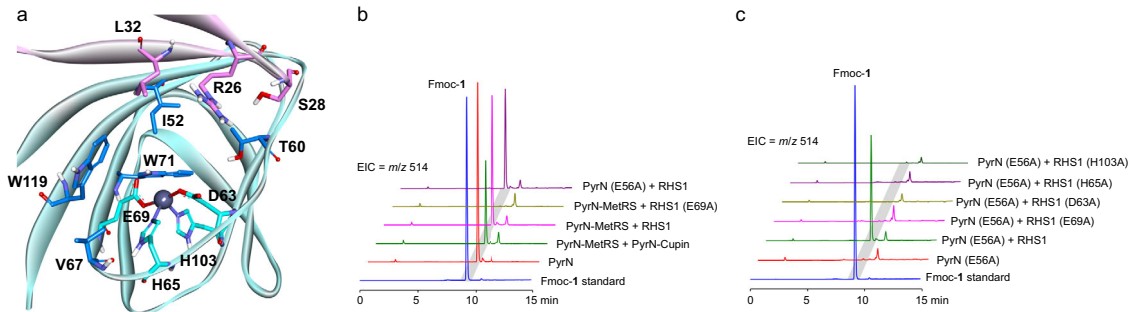

**Fig. 4 Identification of RHS1 as a zinc-binding cupin enzyme catalyzes N-N bond formation. a** The putative active site of RHS1 (reproduced from PDB: 5UQP) with the zinc ion presented as a gray sphere. **b**, **c** In vitro biochemical assays of RHS1 and its variants.

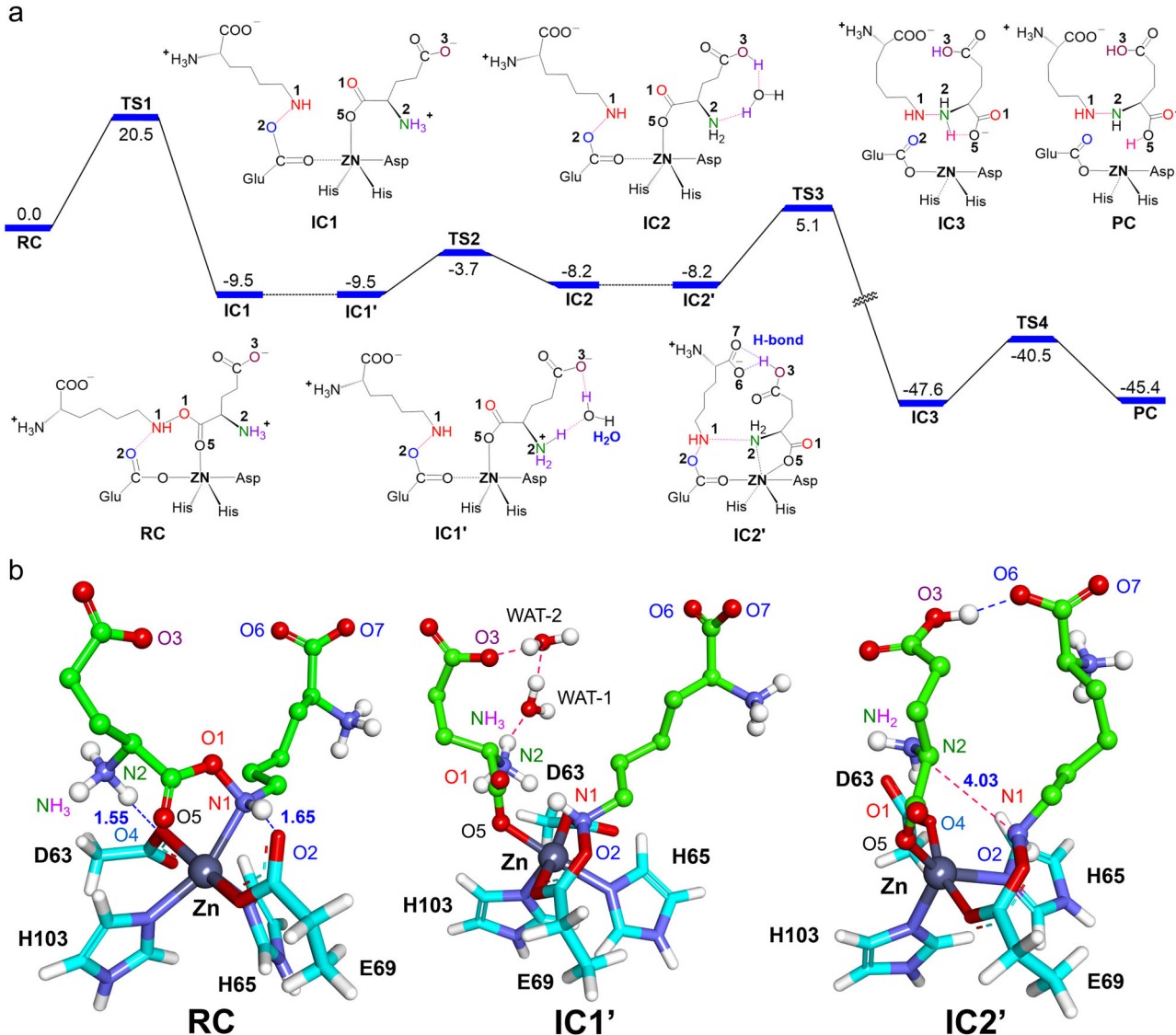

**Fig. 5 QM/MM calculated mechanisms (with energies in kcal/mol) for the RHS1-catalyzed N-N bond formation reactions. a**. QM/MM calculated energy profile for the RHS1-catalyzed N-N bond formation. **b** QM/MM optimized structures of RC, IC1′, and IC2′. Key residues (Asp63, His65, Glu69, and His103) coordinated with the Zn center are labeled. Note: the QM/MM optimized structures of other intermediates displayed in (**a**) can be found in Supplementary Fig. 18.

Starting from RC, our QM/MM calculations show that the N1-O1 cleavage in **4** is coupled with the N1-O2 bond formation between the cleaved N1-containing fragment and the carboxyl group of Glu69 (Fig. 5a). This step experiences the barrier of 20.5 kcal/mol (16.9 kcal/mol in QM simulations in Supplementary Fig. 17a, b) and leads to the formation of N1-O1 cleaved intermediate IC1 (Fig. 5a and Supplementary Fig. 18). In order to initiate the following N-N coupling reaction, the $NH_3$ group of **4** has to lose at least one proton and thus being deprotonated. Starting from IC1, we have investigated three possible reaction pathways for this deprotonation step, including the transfer of proton to: (I) carboxyl O4 of Asp63 (Supplementary Fig. 19), (II) carboxyl O3 of a substrate without bridging waters (Supplementary Fig. 20), and (III) carboxyl O3 of the substrate with the help of bridging waters (Fig. 5a). Among three pathways, we found that the two proton transfer pathways (I) and (II) are unfavorable thermodynamically, while the only feasible pathway is the proton transfer to O3 assisted by two bridging waters. As such, two water molecules and the terminal carboxyl group of the substrate were further included in the QM region (IC1′) (Supplementary

Fig. 16b). For such water-assisted deprotonation of $NH_3$ via TS2, the reaction involves a small barrier of 5.8 kcal/mol (IC1′ → TS2 in Fig. 5a), leading to the intermediate IC2. Our QM model calculations further confirm the vital roles of water molecules in facilitating proton transfer from amino to the carboxyl group (Supplementary Fig. 17a, c).

Staring from IC2, we found the direct N-N coupling is highly unfavorable kinetically. This is mostly because the $NH_2$ group is relatively far away from N1-H and such conformation is highly unfavorable for N-N coupling. As such, MD simulations was carried out to identify if IC2 can experience some conformational change to facilitate the following N-N coupling (IC2 → IC2′ in Fig. 5a). Interestingly, we found that the protonated carboxyl group (O3 site in Fig. 5b) can approach the deprotonated carboxyl group (O6 site in Fig. 5b) and thus forms a persistent H-bond between each other during the MD simulations (Supplementary Fig. 21). We also docked product **1** into RHS1 and found that **1** took a curly binding conformation that is quite similar to IC2′ (Supplementary Fig. 22). All these findings further support the rationality of such conformational change (IC2 →

IC2′ in Fig. 5a). Moreover, we found that this conformational change could trigger the rotation of the N2-H group toward the N1-H group, which would facilitate the subsequent N-N coupling (Fig. 5a, b). Indeed, the N-N coupling in IC2′ requires a relatively low energy barrier of 13.3 kcal/mol (IC2′ → TS3 in Fig. 5a), leading to the formation of NH-NH$_2$ bond in the intermediate IC3. Starting from IC3, we considered two competing pathways. One involves the proton transfer from NH$_2$ to the coordinated Asp63 (Fig. 5a). This proton transfer process requires a small barrier of 2.5 kcal/mol in QM model calculations (IC3$_{qm}$ → TS4-1$_{qm}$), leading to product **1** (Supplementary Fig. 17a, d). However, QM/MM calculations in the presence of the protein environment show such proton transfer is endothermic by 12.5 kcal/mol and thus unfavorable thermodynamically (Supplementary Fig. 23). Clearly, the protein environment, especially the surrounding H-bonding networks of Asp63 can diminish the basicity of Asp63 for the proton-accepting. It is also worth mentioning that this Asp residue is not strictly conserved among the homologs (the corresponding residue in PyrN is His50). In the alternative mechanistic route, the formation of product **1** is mediated by the proton transfer to adjacent substrate carboxyl group, which involves a slight barrier of 7.1 kcal/mol in QM/MM calculations (IC3 → TS4 in Fig. 5a) and 1.9 kcal/mol in QM model calculations (IC3$_{qm}$ → TS4-2$_{qm}$) (Supplementary Fig. 17a and 17d). Overall, our QM/MM calculations provide a full catalytic mechanism for the RHS1-catalyzed N-N bond formation (Fig. 5a). For comparison, we also investigated the water-mediated nonenzymatic N1-O2 cleavage, en route to the N-N coupling product. However, the reaction requires a high barrier of 32.4 kcal/mol and is thus highly unfavorable (Supplementary Fig. 24). We also investigated the underlying mechanism for the nonenzymatic conversion of **4** to **3** using the cluster-continuum model calculations[22,23]. Our calculations show that the overall reaction likely involves three key steps, including the hydration of ester **4**, C-O bond cleavage, and amide bond formation (Supplementary Fig. 25). These calculated results are consistent with our above experimental data, which shows that the conversion of **4** to **3** can proceed nonenzymatically, whereas the production of **1** from **4** only occur in the presence of cupin enzymes.

Our above calculations showed that Glu69 plays a key and unusual role in mediating N-N formation, which is in line with our results from the point-mutation experiments of RHS1 and PyrN, revealing that both the RHS1 (E69A) and PyrN (E56A) variants completely abolished their activities (Fig. 4c and Supplementary Fig. 3). We generated four more RHS1 variants (E69D, E69N, E69Q, and E69L) and then evaluated their protein expression level and in vitro catalytic activity. We found that the E69Q and E69N variants are also inactive, and both the E69D and E69L variants expressed exclusively in the inclusion body, preventing their further activity assays (Supplementary Fig. 26).

**Discovery of new PyrN homologs with alternative amino acid substrate specificity**. To facilitate the application of this hydrazine-forming enzyme family in biocatalysis, we search for new PyrN homologs that could potentially recognize other amino acid substrates, besides L-glutamate and L-glycine. We used Sequence Similarity Network (SSN) tool to analyze the phylogenetic relationships of PyrN homologs in the Uniprot protein database (Supplementary Fig. 27a), which guided our selection of more than ten genes (for the di-domain proteins) or gene pairs (for the standalone MetRS-like proteins and their associated cupin proteins) for gene synthesis and the subsequent catalytic activity evaluation (Supplementary Fig. 27b, c). Besides the enzymes that share the same substrate specificity with Spb40/Tri28, which recognize L-glycine, new homologs that can utilize L-

serine, L-alanine, and L-tyrosine were identified (Supplementary Figs. 28–31). Interestingly, we further found that some of cupin domains/proteins have relaxed substrate specificities, as they could be coupled with a unnative partner, the MetRS-like domain of PyrN, to produce **1** (Supplementary Fig. 32). Considering that **1** was not detected when these cupin domains/proteins were co-expressed with their native MetRS-like domains/proteins, these results suggested that MetRS-like domains/proteins have more strict substrate specificities, and could act as a gatekeeper to determine the outcome of the multi-step reactions. It is also worth to mention that all the cupins we investigated contain an invariant Glu residue that has been suggested to play a critical role in mediating N-N bond formation, based on our above computational studies (Supplementary Fig. 33).

## Discussion

The biosynthetic strategies for constructing N-N bonds in natural products have received great attention over the past decade[13,14,24,25]. Despite that there is an increasing number of dedicated enzymes from different protein families that were discovered to be responsible for the formation of N-N linkages in various natural products, little is known about their catalytic mechanism[26]. In this study, we conducted detailed in vitro characterization of PyrN, a di-domain enzyme that catalyzes hydrazine bond formation in the biosynthetic pathway of anti-viral antibiotic pyrazomycin. Our results revealed that the MetRS-like domain of this enzyme catalyzes an ATP-dependent condensation of L-$N^6$-OH-lysine and L-glutamate to give a highly unstable ester intermediate, which could proceed through a glutamyl-adenylate intermediate, as observed in the first half reaction of canonical aminoacyl-tRNA synthetases (AARSs). However, unlike AARSs, MetRS-like enzymes appear to utilize L-$N^6$-OH-lysine, instead of tRNA, as an amino acid carrier in the second half reaction. In the presence of the zinc-binding cupin domain, the unstable ester product from the MetRS-like domain-catalyzed reaction undergoes redox-neutral, intramolecular arrangement to form an N-N bond containing product. This could be achieved through the zinc-assisted N-O bond cleavage followed by N-N bond formation, as suggested by our computational simulations. Our result thus demonstrated that these zinc-binding cupin domains/proteins represent a novel family of N-N-forming enzymes. Considering that this cupin family enzymes are widespread in bacteria and hydrazine synthases have great potential in biocatalysis, we further performed SSN-guided protein database mining. Such effects have resulted in the successful identification of new homologs of either the full-length PyrN or its MetRS-like and cupin domains, which are able to accept alternative amino acid substrates, expanding the potential application of these interesting biocatalysts.

In conclusion, we reported the catalytic route and the reaction mechanism of N-N bond formation by a family of zinc-binding cupin enzymes, which are widely distributed in the biosynthetic gene clusters of various bacterial specialized metabolites. Our results expanded the current knowledge of enzymatic N-N bond formation and set the stage for the development of novel biocatalysts for the synthesis of useful molecules containing a hydrazine bond.

## Methods

**General methods**. DNA primers (primer sequences are listed in Supplementary Table. 1) were purchased from Tsingke Biological Technology. Reagents were purchased from Sigma-Aldrich, Thermo Fisher Scientific, Cambridge Isotope Laboratories, New England BioLabs, Bio Basic Inc. DNA manipulations in *Escherichia coli* strains were carried out according to standard procedures[27]. Ampicillin (100 μg mL$^{-1}$), apramycin (50 μg mL$^{-1}$), kanamycin (50 μg mL$^{-1}$), and spectinomycin (50 μg mL$^{-1}$) were used for the selection of recombinant *E. coli* strains.

**Isolation and characterization of Fmoc-1.** The *E. coli* strain harboring the vector pCDFDuet-*nbtG* (an $N^6$-lysine hydroxylase gene) and pET28a-*pyrN* was cultivated in M9 medium for protein expression and metabolites production[8]. When the OD600 reached ~0.6, the cultures were supplemented with IPTG at a final concentration of 0.1 mM, along with 0.1% (w/v) of lysine and 0.5% (w/v) of glutamate. The cells were then cultured at 30 °C for another 12 h and the culture broth (400 mL) was centrifuged at 6000 rpm for 10 min. The supernatant was collected and mixed with 4 volumes of ethanol for deproteination. The soluble fraction was concentrated under vacuum, and the resulting residue was dissolved in methanol, dried, and then re-dissolved in water. Subsequent Fmoc chloride (Fmoc-Cl) derivatization was performed as follows: the aqueous solution (15 mL) containing **1** was mixed with 30 mL of acetonitrile, 3 mL 0.2 M borate buffer (pH 8.2), and 150 mg Fmoc-Cl, and incubated at room temperature for 2 h. After being concentrated under vacuum, the residue containing Fmoc-**1** was fractionated on Sephadex LH-20 with MeOH: $H_2O$ (1:1) elution. Metabolites of interest, tracked by LC-MS (EIC = $m/z$ 514, $[M + H]^+$ for Fmoc-**1**), were purified from these fractions containing Fmoc-**1** by reversed-phase semi-preparative HPLC (YMC-Triart C18, 5 μm, 10 mm ID × 250 mm). The $^1H$- and $^{13}C$-and 2D NMR spectra were recorded on a Bruker AV-600 MHz spectrometer using methanol-$d_4$ as the solvent. For isolation and characterization of Fmoc-**1** labeled with $^{15}N$, 0.1% (w/v) of L-ε-$^{15}N$-lysine and 0.5% (w/v) of and L-$^{15}N$-glutamate were fed to the *E. coli* strain expressing *nbtG* and *pyrN*. The isolation of Fmoc-$^{15}N_2$-**1** was performed similarly as described above. The $^{15}N$- and $^1H$-$^{15}N$ HMBC NMR spectra were recorded on a Bruker AV-600 MHz spectrometer using $D_2O$ as the solvent.

**Protein expression and purification.** For the construction of protein expression vectors, DNA fragments containing the coding regions of target genes were either amplified by PCR using the primers listed in Supplementary Table. 1, or obtained through gene synthesis at Sangon Biotech (Shanghai, China). These DNA fragments were cloned into the expression vector pET28a or pCDFDuet-1, and the resulting vectors were then transformed into *E. coli* strain BL21 (DE3) for protein expression. Point mutation of wild-type proteins was introduced by using the Q5 Site-Directed Mutagenesis Kit (NEB) and confirmed by DNA sequencing analysis. For protein expression, cells harboring corresponding expression vectors were grown overnight in 5 mL of Luria–Bertani (LB) broth, supplemented with 50 μg mL$^{-1}$ kanamycin (for pET28a-derived vectors) or 50 μg mL$^{-1}$ spectinomycin (for pCDFDuet-1-derived vectors), at 37 °C and 200 rpm. A starting culture (2.5 mL) was then used to inoculate 750 mL of LB broth containing appropriate antibiotics. The culture was grown at 37 °C and 200 rpm to an optical density of 0.6 at 600 nm. Isopropyl β-D-1-thiogalactopyranoside (IPTG) was then added, at the final concentration of 0.1 mM, to induce protein overproduction.

For protein purification, the cells were harvested after 20 h of further incubation at 16 °C, and resuspended in the lysis buffer (buffer components: 300 mM NaCl, 10 mM imidazole, 1 mM DTT, 50 mM Tris-HCl, pH 8.0) and followed by cell disruption by sonication. After the recovery of supernatant by centrifugation (13,000×g for 40 min), His-tagged protein was separated using nickel-nitrilotriacetic acid (Ni-NTA) resins. The resulting resins were first washed with the washing buffer (buffer components: 300 mM NaCl, 50 mM imidazole, 1 mM DTT, 50 mM Tris-HCl, pH 8.0), and the target protein was then eluted with the elution buffer (buffer components: 300 mM NaCl, 250 mM imidazole, 1 mM DTT, 50 mM Tris-HCl, pH 8.0). The purified protein fractions were confirmed by SDS-PAGE analysis, and then followed by dialysis overnight against 1 L of the storage buffer (buffer components: 150 mM NaCl, 10% glycerol, 40 mM Tris-HCl, pH 8.0). The resulting protein samples were concentrated and stored at −80 °C for further use. Protein concentration was determined at OD280 using a UV-Vis spectrometer, with the extinction coefficients of each protein calculated using the online ProtParam tool (https://web.expasy.org/protparam/). For inductively coupled plasma-mass spectrometry (ICP-MS) analysis, purified protein samples were concentrated to ~200 μM and treated with nitric acid (65%) to release all metal ions before testing.

**In vitro biochemical assays and product analysis by LC-MS.** For the in vitro assays of PyrN (or its variants), the reaction mixture (50 μl) contained 3 mM L-$N^6$-OH-Lys (**2**), 25 μM PyrN (or its variants), 5 mM ATP, 20 mM L-Glu, 10 mM MgCl$_2$, and 1 mM DTT in 40 mM Tris-HCl buffer (pH 8.3). The reaction mixture was incubated for 3 h at 30 °C and then quenched with two volumes of acetonitrile. For the reaction mixtures containing Glu-tRNA, tRNA, or RNase, 10 μL of S30 premix extract (Promega), 0.2 mg/mL *E. coli* total tRNA (Roche), 2 units of RNase inhibitor (Sangon Biotech), or 10 μg/mL RNase A (Sangon Biotech) were included. For the reaction mixtures containing metal-chelating agents 1,10-phenanthroline (OP) or ethylenediaminetetraacetic acid (EDTA), enzymes were preincubated with 5 mM OP (0.5 M stock solution in DMSO) or 5 mM EDTA (0.5 M stock solution, pH 8.0) in Tris-HCl buffer for 3 h at 4 °C, before other components (L-Glu, **2**, ATP, Mg$^{2+}$) were added to initiate the reaction, which was then incubated for another 3 h at 30 °C.

For the in vitro assay of PyrN (E56A) + PyrN (K458A), the reaction mixture (50 μl) contained 3 mM **2**, 25 μM PyrN (E56A), 25 μM PyrN (K458A), 5 mM ATP, 20 mM L-Glu, 10 mM MgCl$_2$, and 1 mM DTT in 40 mM Tris-HCl buffer (pH 8.3). For the in vitro assay of PyrN MetRS and/or PyrN-cupin, the reaction mixture (50 μl) contained 3 mM **2**, 25 μM PyrN MetRS and/or 25 μM PyrN-cupin, 5 mM ATP, 20 mM L-Glu, 10 mM MgCl$_2$, and 1 mM DTT in 40 mM Tris-HCl buffer (pH 8.3).

For the in vitro assay of PyrN (E56A) + RHS1 or its variants, the reaction mixture (50 μl) contains 3 mM **2**, 25 μM PyrN (E56A), 25 μM RHS1 or its variants, 5 mM ATP, 20 mM L-Glu, 10 mM MgCl$_2$, and 1 mM DTT in 40 mM Tris-HCl buffer (pH 8.3).

All the above reaction mixtures were incubated for 3 h at 30 °C and then quenched with two volumes of acetonitrile. The supernatants were then subjected to LC-MS analysis with or without Fmoc-Cl derivatization. LC-MS analysis was carried out with an Agilent 1260 II-6125 apparatus, using an Agilent Elipse XDB-C18 column (5 μm, 4.6 mm ID × 250 mm). Two elution methods were used for the analysis of in vitro reaction mixtures. For the detection of molecules after Fmoc-derivatization, elution was performed at 1 mL min$^{-1}$ with a mobile-phase mixture consisting of a linear gradient of water and acetonitrile ((v/v): 85:15, 0–20 min; 5:95, 20–25 min), both of which contain 0.05% (v/v) formic acid (detection wavelength: 263 nm). For the detection of molecules without derivatization, elution was performed at 0.5 mL min$^{-1}$ with a mobile-phase mixture consisting of a linear gradient of water and acetonitrile ((v/v): 98:2, 0–8 min; 98:2, 8–15 min; 5:95, 15–20 min), both of which contain 0.05% (v/v) formic acid (detection wavelength: 210 and 254 nm). LC-HR-ESI-MS was performed similarly on waters UPLC (Waters Corp., USA) coupled with an AB TripleTOF 5600plus mass spectrometer system (AB SCIEX, USA).

**Enzymatic synthesis of L-$N^6$-$^{18}$OH-Lys and L-$N^6$-OH-$^{13}C_6$-Lys.** For the preparation of L-$N^6$-$^{18}$OH-Lys ($^{18}$O-**2**), a solution (3 mL) consisting of 2 mM L-Lys, 2 mM NADH in 40 mM Tris-HCl buffer (pH 8.0), and a solution (3 mL) consisting of 10 μM PyrM ($N^6$-lysine hydroxylase) in 40 mM Tris-HCl buffer (pH 8.0), were thoroughly bubbled with Argon separately to remove molecule oxygen. Subsequently, the $^{18}O_2$ gas was introduced to these sealed bottles and followed by mixing these solutions together to initiate the PyrM-catalyzed reaction. This reaction mixture was monitored by LC-MS analysis to evaluate the conversion of L-Lys to $^{18}$O-**2**, and the enzyme was removed through ultracentrifugation using Amicon Ultra-4 mL centrifugal filters (Millipore, 3,000 MWCO), when L-Lys completely converts to $^{18}$O-**2**. The filtrate solution was then directly used for subsequent experiments without further purification.

For the preparation of L-$N^6$-OH-$^{13}C_6$-Lys ($^{13}C_6$-**2**), the reaction mixture contains 1 mM L-$^{13}C_6$-Lys, 1 mM NADH, and 5 μM PyrM in 40 mM Tris-HCl buffer (pH 8.0). This reaction mixture was incubated for 3 h at 30 °C and then quenched through ultracentrifugation. The filtrate solution containing $^{13}C_6$-**2** was directly used for subsequent experiments without further purification.

**System setup for computational studies.** The initial structure of the enzyme was prepared on the basis of the determined crystal structure of the cupin protein from *Rhodococcus jostii* RHA1 (PDB code: 5UQP, determined at a resolution of 2.4 Å), was retrieved from the Brookhaven Protein Data Bank (http://www.rcsb.org/pdb). Here, we assigned the protonation states of titratable residues (His, Glu, Asp) based on pKa values from the PROPKA software[28] in combination with a careful visual inspection of local hydrogen-bonded networks. Thereafter, the ester intermediate substrate **4** was docked into the generated pocket of cupin protein using the AutoDock Vina[29] tool in Chimera[30] and the lowest binding energy conformer was selected to make further molecular dynamics (MD) simulation.

**Classical MD simulations.** The Amber ff14SB[31] force field was selected for treating the amino acid residues of the cupin protein, while the AMBER force field (GAFF)[32] was employed for the substrate. Besides, the RESP calculations[33] at the HF/6-31 G* level of theory was been used to define the partial atomic charges of the substrate. The parmchk utility in AmberTools 18 was used to load the missing parameters of the substrate. Then, we added the sodium ions to the surface of the protein to balance the total charge of the complex systems. Finally, the whole system was solvated in a rectangular box with TIP3P waters, with a minimum distance of 15 Å from the protein surface. After the setup of the system, it was totally minimized by the combined steepest descent and conjugate gradient methods. The system was gently annealed from 10 to 300 K under a canonical ensemble for 50 ps with a small restraint of 15 kcal/mol/Å on the protein. In order to get a uniform density, 1 ns of density equilibration was then performed under the NPT ensemble at the target temperature of 300 K and the pressure of 1.0 atm. Afterward, we removed all the restraints on the protein and further equilibrated the system for 10 ns under the NPT ensemble to get the stable temperature and pressure. At last, we performed a productive MD simulation under the NPT ensemble for 50 ns. During the MD, the covalent bonds containing hydrogen were constrained using the SHAKE, and the integration step of 2 fs was used. All the MD process were conducted by the Amber 18 package[34].

**QM calculation.** The QM model is comprised of the substrate, Zn cofactor, as well as Zn-coordinated residues, including Asp63, His65, Glu69, and His103. All QM model calculations were performed with the Gaussian 16 software[35]. The geometries of interested species were fully optimized in conjunction with the SMD[36] continuum solvation model at the B3LYP/def2-SVP level of theory. B3LYP was proven to be a successful functional for studying zinc-based metalloenzymes[37–39].

The energies were further refined with the larger basis set def2-TZVP for all atoms. As for the nonenzymatic reaction, the species with hydrated cluster model[22,23] were optimized in conjunction with the SMD continuum solvation model at the BMK/6-31 G(d) level of theory. The energies were further refined with the larger basis set 6-311 + +G(d,p) for all atoms. The dispersion energies were included in both optimizations and single-point energy calculations. The OriginPro Learning Edition (https://www.originlab.com/OriginProLearning.aspx) was used to process the data from MD simulation, QM/MM and QM scan.

**QM/MM calculation**. For the subsequent QM/MM calculations, we selected the representative snapshot from each classical MD trajectory. All the QM/MM calculations were performed by the ChemShell,[40] in which the turbomole[41] is invoked for the QM region while the DL_POLY[42] is used for the MM region. Besides, the electronic embedding scheme[43] was employed to account for the polarizing effect of the enzyme environment on the QM region, while the hydrogen link atoms with the charge-shift model was used to deal with the QM/MM boundary. Here, the QM region was studied with the hybrid B3LYP[44,45] density functional, which has been proven to be reliable for the simulation of zinc-containing metalloenzymes[37,39,46,47]. In this study, the double-ζ basis set def2-SVP were used for geometry optimization, while the energies were corrected with the larger basis set def2-TZVP for all the QM region atoms. The dispersion corrections with Grimme's D3 method[48] were added in all QM calculations. Similar to QM calculations, the QM region in the QM/MM model included the Zn atom, the substrate, the coordinated Asp63, His65, Glu69, and His103 (See Supplementary Fig. 16 for the selection of QM region). For the transition states (TSs) optimizations, we firstly performed the relaxed potential energy surface (PES) scanning. Then, the located highest point of PES was subject to the full TS optimizations using the DL-FIND code[49].

**Characterization of PyrN homologs selected from Uniprot protein database mining**. The genes or gene pairs, obtained through SSN-guided database mining (Supplementary Fig. 27), were subjected to codon optimization and gene synthesis (at Sangon Biotech Co., Ltd). For fused di-domain enzyme genes, they were cloned into pET28a vector through the NdeI/XhoI sites, and co-introduced into the E. coli strain BL21(DE3) with pCDFDuet-nbtG (an N[6]-lysine hydroxylase gene) for protein expression and metabolite production. For standalone cupin and MetRS-like proteins, the MetRS-like genes were cloned into pET28a through the NdeI/XhoI sites, and the cupins were cloned into the NdeI/XhoI sites of pCDFDuet-nbtG to afford pCDFDuet-nbtG-cupin vectors. The two vectors (pET28a-metRS and pCDFDuet-nbtG-cupin) were then co-introduced into the E. coli for protein expression and metabolite production. For LC-MS analysis, the culture broth supernatants of each strain were first mixed with two volumes of acetonitrile and followed by Fmoc chloride derivatization. The MS signals corresponding to the twenty Fmoc-lysine-AAs (amino acids) conjugates were searched for all the samples, to determine the substrate specificities of synthesized enzymes.

**Reporting summary**. Further information on research design is available in the Nature Research Reporting Summary linked to this article.

## Data availability

The data generated in this study are provided in this Article, Supplementary Information, and Source data files. A reporting summary for this article is available as a Supplementary Information file. All other relevant data are available from the corresponding author upon request. Source data are provided with this paper.

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

## Acknowledgements

This work was supported by the National Natural Science Foundation of China (31872625 and 32122005) and the Zhejiang Provincial Natural Science Foundation (LR19C010001) to Y.-L.D., and the National Key R&D Program of China (2019YFA0906400) to B.W. We thank Zhiwei Ge (Analysis Center of Agrobiology and Environmental Sciences, Zhejiang University) for the assistance of LC-HR-MS/MS data collection.

## Author contributions

G.Z. and Y.-L.D. carried out biochemical, bioinformatics and synthetic work, and performed structural elucidation. W.P. and B.W. performed computational studies on the reaction mechanism. K.S. and J.S. participated in the compound isolation. X.L. carried out NMR spectra collection on the isotope-labeled compounds. G.Z., W.P., B.W., and Y.-L.D. designed the study, analyzed the results, and wrote the paper.

## Competing interests

The authors declare no competing interests.
