## [Peer Review File · Nature Communications]

REVIEWER COMMENTS

Reviewer #1 (Remarks to the Author):

This manuscript describes a detailed analysis of the N-N bond-synthesizing enzyme, PyrN, by mainly using in vitro analysis of the recombinant enzyme. The authors detected N-N bond formation activity using the recombinant enzyme using glutamate and N-hydroxylysine as substrates. PyrN and its homolog, Spd40 were shown to catalyze N-N bond formation in vivo but their function could not be reconstituted in the previous study. Therefore, this result has significant importance. Furthermore, they analyzed the function of cupin and MetRS domains of PyrN individually by using PyrN variants and truncated enzymes. In addition, they analyzed the products using isotope-labeled substrates carefully. These results clearly indicated that the MetRS domain catalyzes the synthesis of ester intermediate (4) using glutamate and N-hydroxylysine via AMPylation. Furthermore, they showed that 4 is unstable and rapidly isomerized to N-glutamyl-N6-hydroxy-lysine in the absence of the active cupin domain. The cupin domain was shown to catalyze the isomerization of 4 to synthesize the N-N bond-containing compound (1). Next, they showed that another cupin family enzyme, RHS1 whose structure was solved previously, can also catalyze the same isomerization as PyrN to synthesize an N-N bond. Based on the structure, they carried out site-directed mutagenesis, docking modeling, MD and QM calculation. As a result, they proposed the reaction mechanism of the isomerization reaction in which Glu69 plays an important role by capturing the cleaved intermediate. Finally, they analyzed several PyrN homologs discovered from the database and analyzed them in vivo and in vitro. As a result, they discovered several PyrN homologs with different substrate specificities resulting in N-N bond-containing compounds with different amino acids instead of the Glu residue. Most of the experiments seemed to be carried out carefully and most of the results are convincing enough to support their conclusion. Since there are still only few information for the mechanism of N-N bond synthesizing enzymes, this manuscript is important for the field of natural products and enzymatic chemistry.

I suggest providing other E69 substituted variants for in vitro analysis in addition to E69A (eg, E69D, N, Q, L) because the E69A also affects the size of the cavity.

Please also discuss the amino acid residues which might be important for interacting with the gamma carboxylic acid of Glu, alpha amine and alpha carboxylic acid of Lys in the model of RHS1.

Please add more descriptions for the QM calculation in the material and methods. Please describe which residues were used for QM layer.

For deprotonation of NH₂, via carboxylic acid of Glu, is there any amino acid residue that might be interacting with Glu to stimulate this process?

How the two water molecules come from in this deprotonation? I could not understand why the authors give these two water molecules. Are these residues stabilized by the protein? Or observed in the crystal structure? If it is not, I am speculative about this process.

Please also provide the energy diagrams of the other steps in the supplementary figure.

I wonder the authors can try to soak the product (1) into the crystal of RHS1 to solve the structure of the complex of RHS1 with the product. Because the condition for crystallization condition for RHS1 should be available, the authors can at least try this. The result should strengthen their hypothesis.

Please also confirm the native function of RHS1 using heterologous expression using MetRS-like domain encoded near its gene. It is useful to confirm the native substrate of it.

Please describe the existence of lysine N-hydroxylases in supplementary figure 18b.

I am interested in the substrate specificities of both MetRS and cupin domains described in Supplementary Figure 18. I assume that MetRS domains have high substrate specificities while

cupin domains may have rather promiscuous substrate specificities. Can authors elucidate this by shuffling the cupin domains in the heterologous expression system?

Authors also should confirm that the isomerization of 4 to 3 and difficulty of 4 to 1 using QM calculation without enzyme?

Multiple grammatical mistakes were found throughout the manuscript. Please check the manuscript carefully. Some of them are listed below.

Line 45, "nonproteinic" should be "nonproteinogenic"

Line 268, one of "that" should be removed.

Line 280, "highly" should be "high"

Line 431, "nitrate acid" should be "nitric acid"

Reviewer #2 (Remarks to the Author):

In this manuscript Zhao et al. show the catalytic process of a stepwise N-N bond formation route involving an esterification followed by rearrangement steps. This paper adds to our knowledge of N-N bond biosynthesis and provides a chance to find new natural products containing N-N linkages. In general the paper is worth publishing but the following issues need to be resolved first:

1. The authors need to show the production of 3 by solely the MetRS-like domain (PyrN-MetRS) to exclude the possible catalytic role of other residues of PyrN-cupin in the proposed first step. Similarly, the authors also have to present the assays with the co-incubation of PyrN-MetRS and RHS1. The currently shown assays employing E56A are inadequate to support the full role of RHS1 due to the presence of almost all residues of the PyrN-cupin.
2. In Fig. 2a, the assay without tRNA should be included and the RNase treated assays can be moved to supplementary material.
3. The point mutations of the proposed metal-binding residues caused partial or full loss of zinc ion binding. For these mutants, the authors should try the supplementation of exogenous zinc to check if activity can be restored.
4. The assays in this paper contain 10 mM MgCl₂, which can bind EDTA via chelation. It therefore needs to be explained why the authors only treat the assays using 5 mM EDTA?
5. Writing mistakes: the "Spd40" (line 63, 80, 127, 211, and 335) and "s56-1" (line 63 and 81) should be "Spb40" and "s56-p1" according to reference 5, respectively.

Reviewer #3 (Remarks to the Author):

The authors describe a mechanistic proposal that is supported largely on the basis of computational results for the formation of N-N bonds by a family of cupin enzymes. The biochemical results follow closely the work of Nishiyama (JACS 2018 9083) and hence the impact and novelty rests in the mechanistic study.

The general mechanism shown in Figure 3 is a 5-endo-tet. This is almost certainly not the case and no data (experimental or computational) has been provided to suggest or support this.

Additionally, the mechanistic study has a flaw that invalidates the general pathway. Namely, the authors have not performed accurate transition state calculations. The only data provided is that in

Sup. Figure 17 where TS1 was located. It was not confirmed that this is the transition state, and in fact this method of locating a transition state is not expected to find it. Instead the authors need to locate the transition states and determine their energies. The coordinates of these TS's need to be provided and the level of theory needs to be provided and benchmarked (none of these things have been done).

Insufficient characterization detail is provided for the new compounds, and if a revision were to be considered against my recommendation, I could outline those issues.

Point-to-point responses

Reviewer #1

REVIEWER COMMENTS: This manuscript describes a detailed analysis of the N-N bond-synthesizing enzyme, PyrN, by mainly using in vitro analysis of the recombinant enzyme. The authors detected N-N bond formation activity using the recombinant enzyme using glutamate and N-hydroxylysine as substrates. PyrN and its homolog, Spd40 were shown to catalyze N-N bond formation in vivo but their function could not be reconstituted in the previous study. Therefore, this result has significant importance. Furthermore, they analyzed the function of cupin and MetRS domains of PyrN individually by using PyrN variants and truncated enzymes. In addition, they analyzed the products using isotope-labeled substrates carefully. These results clearly indicated that the MetRS domain catalyzes the synthesis of ester intermediate (**4**) using glutamate and N-hydroxylysine via AMPylation. Furthermore, they showed that **4** is unstable and rapidly isomerized to N-glutamyl-N6-hydroxy-lysine in the absence of the active cupin domain.

The cupin domain was shown to catalyze the isomerization of **4** to synthesize the N-N bond-containing compound (**1**). Next, they showed that another cupin family enzyme, RHS1 whose structure was solved previously, can also catalyze the same isomerization as PyrN to synthesize an N-N bond. Based on the structure, they carried out site-directed mutagenesis, docking modeling, MD and QM calculation. As a result, they proposed the reaction mechanism of the isomerization reaction in which Glu69 plays an important role by capturing the cleaved intermediate. Finally, they analyzed several PyrN homologs discovered from the database and analyzed them in vivo and in vitro. As a result, they discovered several PyrN homologs with different substrate specificities resulting in N-N bond-containing compounds with different amino acids instead of the Glu residue. Most of the experiments seemed to be carried out carefully and most of the results are convincing enough to support their conclusion. Since there are still only few information for the mechanism of N-N bond synthesizing enzymes, this manuscript is important for the field of natural products and enzymatic chemistry.

Response: Thank you for your positive comments.

REVIEWER COMMENTS: I suggest providing other E69 substituted variants for in vitro analysis in addition to E69A (eg, E69D, N, Q, L) because the E69A also affects the size of the cavity.

Response: Thank you for your useful suggestion. We have generated four more E69 substituted variants (E69D, E69N, E69Q, E69L), and evaluated their protein expression level and in vitro catalytic activity. We found that the E69Q and E69N variants were also inactive, but surprisingly, both the E69D and E69L variants expressed exclusively in the inclusion body, preventing their further activity assays (Supplementary Fig. 26).

REVIEWER COMMENTS: Please also discuss the amino acid residues which might be important for interacting with the gamma carboxylic acid of Glu, alpha amine and alpha carboxylic acid of Lys in the model of RHS1.

Response: Thank you for your suggestion. We have added relevant interactions in Supplementary Fig. 14a, and described these interactions in detail in the Figure legend.

REVIEWER COMMENTS: Please add more descriptions for the QM calculation in the material and methods. Please describe which residues were used for QM layer.

Response: Following the suggestion of the referee, we have revised the description for the selection of QM regions in QM method section (Supplementary Fig. 16). In addition, we have added a schematic picture for the selection of QM region of QM/MM in Supplementary Fig. 16.

REVIEWER COMMENTS: For deprotonation of NH₂, via carboxylic acid of Glu, is there any amino acid residue that might be interacting with Glu to stimulate this process? How the two water molecules come from in this deprotonation? I could not understand why the authors give these two water molecules. Are these residues stabilized by the protein? Or observed in the crystal structure? If it is not, I am speculative about this process.

Response: For deprotonation of NH₂, the QM/MM calculations in the presence of the protein environment show such proton transfer is endothermic by 12.5 kcal/mol and thus unfavorable thermodynamically. Here, we found that the protein environment, especially the surrounding H-bonding networks of Asp63 (e.g. carboxylic acid of Asp63 and amino group of substrate, the water-mediated hydrogen bond between hydroxyl group in Thr60 and carboxylic acid of Asp63) can diminish the basicity of Asp63 for the proton-accepting. However, the proton transfer from amino group to adjacent carboxyl group was demonstrated to be quite favorable in both the QM/MM and QM calculations. For detailed result, please check in Main text Line 361~373.

The two water molecules cannot be observed in the crystal structure since the two waters are mediated by the bound substrate while in the crystal structure the substrate is absent. However, our long-term MD simulation indicated that two waters can penetrate into the active site and thus mediate a persistent H-bonding network between the positively charged substrate-NH₃⁺ group and the CO₂⁻ group (Figure 5b, Supplementary Fig. 14d and 18). Such water-mediated proton channel is stabilized by the substrate itself, while has little interaction with protein. Indeed, the water-mediated proton transfer is ubiquitous in biological systems. For related discussion, please check in Main text Line 334~346.

REVIEWER COMMENTS: Please also provide the energy diagrams of the other steps in the supplementary figure.

Response: We have provided energy diagrams for each reaction step of QM calculation in Supplementary Fig. 17b-d.

REVIEWER COMMENTS: I wonder the authors can try to soak the product (**1**) into the crystal of RHS1 to solve the structure of the complex of RHS1 with the product. Because the condition for crystallization condition for RHS1 should be available, the authors can at least try this. The result should strengthen their hypothesis.

***Response:** Thank you for your suggestion. We have tried to repeat the crystallization condition (0.2 M Potassium Iodide, 20% (w/v) PEG 3350) for RHS1. However, we did not get RHS1 crystals using this buffer condition. A molecular docking model of RHS1 with the product **1** is shown in Supplementary Fig. 22. Crystallization of other cupin proteins (listed in Supplementary Fig. 27b) are currently underway, hope we could get a co-crystal structure of one of these cupins with the corresponding product, and publish the results in the near future.*

REVIEWER COMMENTS: Please also confirm the native function of RHS1 using heterologous expression using MetRS-like domain encoded near its gene. It is useful to confirm the native substrate of it.

***Response:** Thank you for your suggestion. We found that this MetRS-like protein expressed exclusively in insoluble form in *E. coli* system. However, we were able to overexpress this MetRS-like gene, together with its associated cupin and lysine N⁶-hydroxylase genes in the original *Rhodococcus* strain (Supplementary Fig. 10b). LC-MS analysis of the culture supernatant of this engineered *Rhodococcus* strain revealed the production of **1**, supporting that this MetRS-like domain protein naturally utilize L-Glu.*

REVIEWER COMMENTS: Please describe the existence of lysine N-hydroxylases in supplementary figure 18b.

***Response:** we have added this information in the new Supplementary Fig. 27c, showing that all the genomic regions harboring genes/gene pairs selected for synthesis, also contain N⁶-lysine hydroxylase genes.*

REVIEWER COMMENTS: I am interested in the substrate specificities of both MetRS and cupin domains described in Supplementary Figure 18. I assume that MetRS domains have high substrate specificities while cupin domains may have rather promiscuous substrate specificities. Can authors elucidate this by shuffling the cupin domains in the heterologous expression system?

***Response:** Thank you for the insightful suggestion. We have tested the substrate specificities of selected cupins in an *E. coli* system expressing N⁶-lysine hydroxylase and PyrN MetRS-like domain (Supplementary Fig. 32). We found that the cupins that naturally use amino acids with relatively larger side chains (L-tyrosine and L-serine) are also able to accept the unstable ester intermediate **4** to produce **1**, supporting the promiscuous substrate specificities of cupin domains/proteins (also see Main text Line 413-419).*

REVIEWER COMMENTS: Authors also should confirm that the isomerization of **4** to **3** and difficulty of **4** to **1** using QM calculation without enzyme?

*Response: Following the suggestion of the referee, we further investigated the mechanism of the non-enzymatic transformation of **4** into the amide product **3** using the cluster-continuum model calculations. Our calculations show that the non-enzymatic transformation of **4** into the amide product **3** involves an overall barrier of 17.7 kcal/mol and thus is favorable kinetically (supplementary Fig. 25, the detailed description of mechanism can be found in the Main text Line 372-376 and page 37 in SI). In addition, we also investigated the non-enzymatic NI-O2 cleavage, en route to the product **1**, requires a high barrier of 32.4 kcal/mol and thus highly unfavorable kinetically (Supplementary Fig. 24, the detailed description of mechanism can be found in the Main text Line 369-372).*

REVIEWER COMMENTS: Multiple grammatical mistakes were found throughout the manuscript. Please check the manuscript carefully. Some of them are listed below.

Line 45, “nonproteinic” should be “nonproteinogenic”

Line 268, one of “that” should be removed.

Line 280, “highly” should be “high”

Line 431, “nitrate acid” should be “nitric acid”

Response: Thank you for pointing out these typos, we have fixed them.

Reviewer #2:

REVIEWER COMMENTS: In this manuscript Zhao et al. show the catalytic process of a stepwise N-N bond formation route involving an esterification followed by rearrangement steps. This paper adds to our knowledge of N-N bond biosynthesis and provides a chance to find new natural products containing N-N linkages. In general the paper is worth publishing but the following issues need to be resolved first.

Response: Thank you for your positive comments.

REVIEWER COMMENTS: The authors need to show the production of **3** by solely the MetRS-like domain (PyrN-MetRS) to exclude the possible catalytic role of other residues of PyrN-cupin in the proposed first step. Similarly, the authors also have to present the assays with the co-incubation of PyrN-MetRS and RHS1. The currently shown assays employing E56A are inadequate to support the full role of RHS1 due to the presence of almost all residues of the PyrN-cupin.

*Response: Thank you for your suggestion. We have prepared new PyrN MetRS-like domain protein by switching the C-terminal His₆-tag to the N-terminus, which improved the activity of this truncated MetRS-like domain (PyrN-MetRS), PyrN (Δ 1-108 aa). We then use this new PyrN-MetRS enzyme to conduct the experiments suggested by the reviewer., The results showed that PyrN-MetRS alone is sufficient for the production of **3** (see Fig. 3a in Main text), and co-incubation of PyrN-MetRS with RHS1 produced **1**, supporting the roles of PyrN MetRS-like domain and RHS1 (see Fig. 4b in Main text).*

REVIEWER COMMENTS: In Fig. 2a, the assay without tRNA should be included and the RNase treated assays can be moved to supplementary material.

Response: Thank you for your suggestion. We have made these changes as suggested.

REVIEWER COMMENTS: The point mutations of the proposed metal-binding residues caused partial or full loss of zinc ion binding. For these mutants, the authors should try the supplementation of exogenous zinc to check if activity can be restored.

Response: Thank you for your useful suggestion. We have performed this experiment, and the results revealed that supplementation of exogenous zinc cannot restore the activities of these variants (Supplementary Fig. 13b).

REVIEWER COMMENTS: The assays in this paper contain 10 mM MgCl₂, which can bind EDTA via chelation. It therefore needs to be explained why the authors only treat the assays using 5 mM EDTA?

Response: We are sorry that we forgot to add the method details of this experiment and have added this information in the Material and Methods part (See Main text Line 529-532). In these assays, enzymes were first pre-incubated with OP or EDTA in Tris-HCl buffer for 3h at 4°C, before other components (L-Glu, 2, ATP, Mg²⁺) were added to initiate the reaction, which was then incubated for another 3 h at 30°C.

REVIEWER COMMENTS: Writing mistakes: the “Spd40” (line 63, 80, 127, 211, and 335) and “s56-1” (line 63 and 81) should be “Spb40” and “s56-p1” according to reference 5, respectively.

Response: Thank you for pointing out these typos, we have fixed them.

Reviewer #3 (Remarks to the Author)

REVIEWER COMMENTS: The authors describe a mechanistic proposal that is supported largely on the basis of computational results for the formation of N-N bonds by a family of cupin enzymes. The biochemical results follow closely the work of Nishiyama (JACS 2018 9083) and hence the impact and novelty rests in the mechanistic study.

Response: The work by Nishiyama et al beautifully identified the function of this di-domain protein family as N-N bond forming enzymes, by using in vivo studies. However, due to the lack of in vitro data, the details about this highly unusual transformation, including the catalytic roles of each domain, the cofactor/metal ion involved, and the reaction intermediate(s) and catalytic mechanism, remained largely unknown. Here, we successfully reconstituted the PyrN-catalyzed reaction in vitro, and performed a detailed analysis of this type of N-N bond forming reaction through both in vitro biochemical and computational studies. we believe that our results expanded the current knowledge of enzymatic N-N bond formation and natural product biosynthesis.

REVIEWER COMMENTS: The general mechanism shown in Figure 3 is a 5-endo-tet. This is almost certainly not the case and no data (experimental or computational) has been provided to suggest or support this.

Response: We are sorry about the misunderstanding. We were only intended to show the atoms connected in the final product 3/1, after the intramolecular arrangement. We have made the correction by removing the arrows (see the new Fig. 3e in Main text).

REVIEWER COMMENTS: Additionally, the mechanistic study has a flaw that invalidates the general pathway. Namely, the authors have not performed accurate transition state calculations. The only data provided is that in Sup. Figure 17 where TS1 was located. It was not confirmed that this is the transition state, and in fact this method of locating a transition state is not expected to find it. Instead the authors need to locate the transition states and determine their energies. The coordinates of these TS's need to be provided and the level of theory needs to be provided and benchmarked (none of these things have been done).

*Response: We thank the referee for his/her constructive comments and insightful suggestions. Following the referee's suggestion, the detailed energy diagrams, the frequency data along with the coordinates of the transition state for each step of QM calculation have been added in this manuscript. Please check the Supplementary Fig. 17, 24 & 25 and the Supplementary data 1. In addition, we have re-examined all reaction steps with QM/MM calculations, in which all the transition states were located by relaxed potential energy surface scans followed by full TS optimizations using the DL-FIND code. Our QM-predicted mechanism is consistent with QM/MM calculations, confirming the calculations are reliable enough. Following the suggestions of the referee, we have tested various DFT functionals for the initial N1-O1 cleavage step (see the Table below). We found that B3PW91 and M06L predict similar barriers as B3LYP, while for M06 and wb97xd, the predicted barriers are higher. For BP86 and TPSS, the predicted barriers are much lower than that from B3LYP. Indeed, B3LYP has been extensively used in studying the Zn-containing enzymes and demonstrated to be quite reliable (Refs, Abdel-Azeim, S. et al. *J. Comput. Chem.* **32**, 3154–3167 (2011); Samanta, P. N. & Das, K. K. *J. Mol. Graph. Model.* **63**, 38–48 (2016); Fu, Y. et al. *Front. Chem.* **9**, 706959 (2021); Sen, A. et al. *J. Phys. Chem. B* **125**, 8814–8826 (2021).)*

Table QM calculated energy barriers (in kcal/mol) for the N1-O1 cleavage with different DFT methods.

Functional	RC _{qm}	TS1 _{qm}	ΔE [‡]
B3LYP	-3147.725138	-3147.698067	17.0
B3PW91	-3147.143063	-3147.111512	19.4
M06	-3146.771690	-3146.735526	22.7
M06L	-3147.433126	-3147.410822	14.6
BP86	-3147.931959	-3147.917821	8.9
wb97xd	-3147.312124	-3147.272453	24.9
TPSS	-3147.877943	-3147.862843	9.5

REVIEWER COMMENTS: Insufficient characterization detail is provided for the new compounds, and if a revision were to be considered against my recommendation, I could outline those issues.

Response: we have added more information about the characterization detail for the new compounds in the Materials and Methods part (see Main text Line 636-651). We realize that we do not have the direct evidence about the N-N linkage in these new products, their structures were suggested based on the fragment pattern and the protein sequence homology of those enzymes to PyrN (We have added this sentence in the figure legend of Supplementary Fig. 29). However, it is clear that these newly-identified enzymes utilize alternative amino acid substrates.

REVIEWERS' COMMENTS

Reviewer #1 (Remarks to the Author):

The revised paper is clearly an improvement over the previous one. The authors have performed heterologous expression of the RHS1 gene and related genes in *Rhodococcus*, which shows the function of this enzyme in vivo. They have also added a description of the experimental results of the QM/MM calculations, which was lacking in the first paper, and described them in more detail. Several additional mutation experiments have been performed, and the results obtained support the authors' hypothesis. They also showed the possibility that the substrate specificity of the enzyme with the cupin domain is relatively loose. The explanation of the experimental results is clearer and more convincing than before. I have some minor suggestions as follows.

Reviewer #2 (Remarks to the Author):

The authors have nicely addressed my concerns and comments. The paper certainly warrants publication.

Reviewer #3 (Remarks to the Author):

The authors have addressed all of my detailed concerns about characterization and the coordinates, and seem to have addressed all of the other referee's concerns. I recommend publication of the revised manuscript.

Line 364-368, the authors may also mention here that the Asp63 is not strictly conserved among the homologs (Supplementary Figure 33).

Line 378-380, Please comment on the consistency of the experimental and calculated results.

Line 46, "nonproteinic amino acid" is not used generally. It should be "Nonproteinogenic amino acids".

Point-to-point responses

Reviewer #1

REVIEWER COMMENTS: The revised paper is clearly an improvement over the previous one. The authors have performed heterologous expression of the RHS1 gene and related genes in *Rhodococcus*, which shows the function of this enzyme in vivo. They have also added a description of the experimental results of the QM/MM calculations, which was lacking in the first paper, and described them in more detail. Several additional mutation experiments have been performed, and the results obtained support the authors' hypothesis. They also showed the possibility that the substrate specificity of the enzyme with the cupin domain is relatively loose. The explanation of the experimental results is clearer and more convincing than before.

Response: Thank you for your positive comments.

I have some minor suggestions as follows.

- Line 364-368, the authors may also mention here that the Asp63 is not strictly conserved among the homologs (Supplementary Figure 33).

Response: we have added this sentence: "It is also worth to mention that this Asp residue is not strictly conserved among the homologs (the corresponding residue in PyrN is His50)." (Line 319-321), as suggested.

- Line 378-380, Please comment on the consistency of the experimental and calculated results.

Response: we have added this sentence: "These calculated results are consistent with our above experimental data, which shows that the conversion of 4 to 3 can proceed non-enzymatically, whereas the production of 1 from 4 only occur in the presence of cupin enzymes." (Line 333-336), as suggested.

- Line 46, "nonproteinic amino acid" is not used generally. It should be "Nonproteinogenic amino acids".

Response: we have changed "nonproteinic" to "nonproteinogenic", as suggested.

Reviewer #2:

REVIEWER COMMENTS: The authors have nicely addressed my concerns and comments. The paper certainly warrants publication.

Response: Thank you for your positive comments.

Reviewer #3 (Remarks to the Author)

REVIEWER COMMENTS: The authors have addressed all of my detailed concerns about characterization and the coordinates, and seem to have addressed all of the other referee's concerns. I recommend publication of the revised manuscript.

Response: Thank you for your positive comments.